# The Effect of Choline and Resistance Training on Strength and Lean Mass in Older Adults

**DOI:** 10.3390/nu15183874

**Published:** 2023-09-06

**Authors:** Chang Woock Lee, Teak V. Lee, Elfego Galvan, Vincent C. W. Chen, Steve Bui, Stephen F. Crouse, James D. Fluckey, Stephen B. Smith, Steven E. Riechman

**Affiliations:** 1Department of Health and Human Performance, Nursing and Counseling, University of Houston-Victoria, Victoria, TX 77901, USA; leec3@uhv.edu; 2Life Sciences Department, Pierce College, Woodland Hills, CA 91367, USA; leetv@piercecollege.edu; 3School of Osteopathic Medicine in Arizona, A.T. Still University, Mesa, AZ 85206, USA; fego.galvan@atsu.edu; 4Department of Integrative Health and Exercise Science, Georgian Court University, Lakewood, NJ 08701, USA; vchen@georgian.edu; 5Department of Health and Human Performance, Utah Tech University, St. George, UT 84770, USA; steve.bui@utahtech.edu; 6Department of Kinesiology and Sport Management, Texas A&M University, College Station, TX 77843, USA; s-crouse@tamu.edu (S.F.C.); jfluckey@tamu.edu (J.D.F.); 7Department of Animal Science, Texas A&M University, College Station, TX 77843, USA

**Keywords:** sarcopenia, exercise, strength, skeletal muscle, choline, nutrition, older population

## Abstract

Choline plays many important roles, including the synthesis of acetylcholine, and may affect muscle responses to exercise. We previously observed correlations between low choline intake and reduced gains in strength and lean mass following a 12-week resistance exercise training (RET) program for older adults. To further explore these findings, we conducted a randomized controlled trial. Three groups of 50-to-69-year-old healthy adults underwent a 12-week RET program (3x/week, 3 sets, 8–12 reps, 70% of maximum strength (1RM)) and submitted >48 diet logs (>4x/week for 12 weeks). Participants’ diets were supplemented with 0.7 mg/kg lean/d (low, n = 13), 2.8 mg/kg lean/d (med, n = 11), or 7.5 mg/kg lean/d (high, n = 13) of choline from egg yolk and protein powder. The ANCOVA tests showed that low choline intake, compared with med or high choline intakes, resulted in significantly diminished gains in composite strength (leg press + chest press 1RM; low, 19.4 ± 8.2%; med, 46.8 ± 8.9%; high, 47.4 ± 8.1%; *p* = 0.034) and thigh-muscle quality (leg press 1RM/thigh lean mass; low, 12.3 ± 9.6%; med/high, 46.4 ± 7.0%; *p* = 0.010) after controlling for lean mass, protein, betaine, and vitamin B_12_. These data suggest that low choline intake may negatively affect strength gains with RET in older adults.

## 1. Introduction

Choline plays crucial roles in several physiological processes, such as neurotransmission and muscle contraction via synthesis of the chemical messenger acetylcholine (ACh), lipid transport via lipoprotein synthesis, and methyl-group metabolism as a precursor to betaine [1,2]. It also supports cell membrane integrity/function as a precursor to phosphatidylcholine (PC), the most abundant phospholipid in all membranes. Choline can be obtained from diet (exogenously) or from de novo PC synthesis (endogenously) [1]. However, the amount of choline produced via de novo synthesis is not sufficient to support the total choline requirement [1,3,4,5,6]. Therefore, the majority of the required choline must be acquired from the diet, and a choline-deficient diet (<10% of Adequate Intake (AI)) has been associated with negative health conditions, including liver/muscle damage, organ dysfunction, atherosclerosis, birth defects, and neurological disorders [7].

Insufficient choline intake can negatively affect exercises because choline is a precursor to ACh, which mediates muscle contraction and force generation [8,9]. Choline may influence exercise performance through its role in maintaining cell membrane integrity. Insufficient choline may contribute to a weakened cell membrane [10], negatively affecting skeletal muscles’ ability to withstand the mechanical stress of exercise. Choline may also affect muscle responses to exercise via betaine, which is important for gene expression/protein synthesis [11,12].

Previously, we have observed that a lower intake of choline was associated with reduced strength and lean mass gains following 12 weeks of resistance exercise training (RET) in 60-to-69-year-old individuals [13]. In that study, subjects’ habitual, food-based choline consumption ranged between ~49% and ~85% of AI, and a positive linear relation was observed between choline intake and percent change in strength within that range [13]. However, it is still unclear whether a higher choline intake (≥AI) positively affects strength and lean mass gains with RET. 

Previous studies regarding choline supplementation and exercise generally reported that choline intake exceeding AI, sufficient to increase blood concentrations of choline, does not positively affect exercise performance [14,15,16,17]. For example, choline supplementation of ~970 mg (~200% AI) prior to cycling did not affect exercise heart rate, ventilation, oxygen consumption, time to exhaustion, and total work output in young athletes [16]. Similarly, 6 g of choline supplementation (>1000% AI) before and during four hours of treadmill-walk, run-to-exhaustion, and squat-to-failure tests did not affect oxygen uptake, heart rate, or time to exhaustion in young male soldiers [17]. However, these studies lacked nutritional control of habitual choline intake, utilized the acute choline supplementation, or examined acute responses or endurance performance variables, leaving the effects of chronic choline intake greater than AI on RET and training responses largely unknown.

The purpose of the present study was to determine the effects of different amounts of choline intakes (approximately 50%, 70%, and 120% of AI) on muscle responses to RET. To maintain consistency with our previous study, we used similar exercise and nutrition control protocols [13]. We supplemented subjects’ habitual choline intake from a standard healthy diet, as recommended by the American Dietetic Association (ADA, now known as the Academy of Nutrition and Dietetics) [18], with choline in the form of egg yolk. Then, we examined the effects of choline on changes in strength, lean mass, and muscle quality with 12 weeks of RET in older males and females in a randomized double-blind placebo-controlled study design. We hypothesized that a lower choline intake would result in less strength and lean mass gains.

## 2. Materials and Methods

### 2.1. Participants

Thirty-seven generally healthy 50-to-69-year-old males and females were recruited via flyers and advertisements in a local newspaper. Smokers and individuals with any of the following health conditions were excluded: hypertension (>160/100 mmHg), cardiac arrhythmias, cancer, hernia, aortic aneurysm, kidney disease, diabetes, lung disease, and blood cholesterol >240 mg/dL or <160 mg/dL or taking cholesterol-lowering medications. Those who participated in one hour or more of RET per week in the previous year were not eligible for participation, and females needed to be postmenopausal for more than two years. The eligible participants were randomly assigned to one of three choline groups in a double-blind manner: zero additional egg yolk (low), one additional egg yolk (medium = med), or three additional egg yolks (high) per day. 

This study was conducted in accordance with the Declaration of Helsinki and approved by Texas A&M University Institutional Review Board. All of the participants provided written informed consent prior to participation in the study. All testing, measurement, and RET were performed in the morning hours in an air-conditioned laboratory. All study data were collected over two years during spring, summer, or fall semesters at Texas A&M University by investigators trained in exercise testing and prescription, as well as experimental data collection.

### 2.2. Orientation

During two weeks of a pre-study orientation period, the participants attended two sessions of nutrition education by a registered dietitian and four sessions of exercise orientation/familiarization (Figure 1). Each nutrition education session lasted for two hours, and the participants learned about proper nutrient intake, calorie/portion control, and study-specific dietary guidelines. They also practiced the use of a nutrition software program (NutriBase, version 7, Client Intake Module (Cybersoft Inc., Phoenix, AZ, USA)) with which they maintained diet logs throughout the study. The exercise orientation provided the participants with information on the benefits of regular exercise and principles of resistance exercise (RE). Correct exercise techniques were explained/demonstrated, and the participants became familiarized with RE by practicing the techniques with light weight initially and gradually increasing the intensity to 40% of their estimated maximum strength (4/10 on the Omnibus-RE Scale (OMNI-RES) ratings of perceived exertion (RPEs)) [19]. The purpose of the exercise orientation was to allow for rapid motor learning while minimizing skeletal muscle adaptations to standardize strength measures, estimate maximum strength (1RM) prior to testing, and reduce the possibility of exercise-induced injury.

### 2.3. Testing

Following the orientation, 1RM, peak power, body composition, resting metabolic rate (RMR), and blood tests were conducted. The 1RMs for all the exercises included in the RET program were measured by gradually increasing exercise weights until the maximum resistance, at which only one repetition was completed with proper form in full range of motion, was reached [19], using Keiser 300 series pneumatic exercise machines (Keiser, Palo Alto, CA, USA). Following a three-minute warm-up on a cycle ergometer (Schwinn Fitness, Inc., Denver, CO, USA) and stretching, participants performed four warm-up repetitions with an exercise weight corresponding to 55% of an estimated 1RM based on RPE on the OMNI-RES. The weight was then increased to 75% of a re-estimated 1RM (based on RPE) to perform only one repetition. After 60 s of rest, the weight was increased again to 90% of a re-estimated 1RM to perform one repetition. Additional attempts for 1RM were made after 60 s of rest until the true 1RM value was obtained in a manner that the total number of 1RM attempts was minimized [19]. The same procedure was performed for all exercises and in the same order for all participants. Based on guidelines of American College of Sports Medicine (ACSM), the composite strength value was calculated using chest press 1RM and leg press 1RM to represent full-body strength [13,20]. The power (force × speed) output for each exercise was also measured during 1RM tests. The participants were instructed to perform the concentric phase of repetitions at their maximal speed, and the Keiser machines calculated the power output for each repetition. The highest value of the power output recorded during 1RM tests was used as peak power for each exercise.

Body composition was assessed by dual-energy X-ray absorptiometry (DEXA), using Lunar Prodigy (General Electric, Fairfield, CT, USA), and RMR was measured with ParvoMedics TrueMax 2400 Metabolic Measurement System (Sandy, UT, USA) in the morning after an overnight fast. Fasted (12 h, overnight) blood samples were collected from antecubital veins, and blood lipid/metabolic panels were run with standard methods at St. Joseph Regional Health Center’s CDC certified laboratory (Bryan, TX, USA). The effects of choline intake on clinical markers of liver/muscle damage and blood lipid profile were examined since choline deficiency is shown to cause liver/muscle damage and perturb lipid metabolism [7]. All tests were repeated at the completion of 12 weeks of RET.

### 2.4. RET

Based on the recommendations from ACSM and American Heart Association (AHA) [21,22,23], the participants performed a full-body RET program 3 times per week (on non-consecutive days) for 12 weeks, using Keiser 300 series exercise machines. The RET program consisted of 10 min of warm-up on a cycle ergometer (Schwinn Fitness, Inc., Denver, CO, USA), five minutes of dynamic stretching, seated chest press, lat pull down, leg press, calf raises, seated leg curls, knee extension, biceps curls, and triceps extension exercises to train major muscle groups. Participants performed three sets of 8–12 repetitions, with resistance set at 70% of 1RM [22]. They were instructed to perform as many repetitions as possible until they reached 12 repetitions or muscle failure on a given set. When a participant was able to complete 12 repetitions on all three sets of an exercise, the weight was increased so that only eight repetitions would be possible during the next exercise session. The rest between each set was 1 min, and the rest between each exercise was 2 min, during which muscle-specific stretching was performed [22]. All the exercise sessions were supervised by exercise physiology graduate students who were trained in exercise testing and prescription, and the participants were instructed to maintain their non-RET physical activities at the pre-study level, but not to perform any additional RET. RET was conducted for 12 weeks to ensure that any strength gains reflected muscle hypertrophy, as well as neural adaptation [24,25].

### 2.5. Nutrition Control

Participants were instructed to consume 50% of total calories from carbohydrates, 30% from fat, 20% from protein, and <10% from saturated fat to meet daily caloric consumption goals, as determined by RMR test. They were also instructed to consume >1.0 g/kg/d of protein, 25–30 g/d of fiber, and <200 mg/d of cholesterol, as recommended by the ADA [18]. Participants were required to maintain 24-h diet logs at least four times per week (three weekdays and one weekend day) during the study period [13,26,27]. Feedback on the diet logs was provided weekly, and adjustments were made as necessary to ensure adherence to the dietary guidelines of the study.

To minimize any potential effect that the variability of protein consumption may have [28], participants consumed protein supplements (0.4 g/kg lean mass, MET-Rx protein [MET-Rx USA Inc., Boca Raton, FL, USA] + egg protein) every 12 h throughout the study period. The supplement also contained different amounts of whole-egg powder, egg-white powder, and peanut oil so that low, med, and high groups were provided with 0.7, 2.8, or 7.5 mg of choline/kg lean/d, respectively, while the same amounts of protein (0.8 g/kg lean/d), carbohydrate (0.9 g/kg lean/d), and fat (0.3 g/kg lean/d) were provided equally for each group.

### 2.6. Thigh-Muscle Quality

From the DEXA scans of each participant, thigh-muscle quality–strength (TMQ-S) was assessed and defined as leg press 1RM (kg)/total thigh lean mass for both lower limbs (kg). Total thigh lean mass was determined through the construction of a four-sided polygon encompassing the entire region of each thigh and combining lean mass of both thighs together (Figure 2). The first line segment of the polygon consisted of one point (point a) inferior to the pubic bone immediately below any flesh as a reference point with the other point (point b) positioned as to obliquely transverse the intertrochanteric crest of the femur bone. The next line segment transected the tibiofemoral joint (c–d), and two more line segments were drawn to enclose the entire thigh tissue (b–c, a–d). Lean mass values located inside the polygon were calculated with DEXA and defined as thigh lean mass. All thigh lean mass measurements were performed by two independent raters. Inter-rater reliability was R^2^ > 0.99, and the coefficient of variation was <1.5%. Deviations from these norms were reanalyzed. Means of the two raters were used for data analyses.

### 2.7. Data Analysis

All statistical analyses were conducted using SAS/STAT software (version 9.4; SAS Institute Inc., Cary, NC, USA). The mean intakes of all nutrients were calculated from the diet logs that were entered into NutriBase software (version 7, Cybersoft Inc., Phoenix, AZ, USA) or direct calculations from the USDA database for choline [29]. The assumption of normal distribution was checked using the Shapiro–Wilk test, and non-normal variables were log transformed for parametric statistical tests. Student’s independent *t*-tests were used to compare means of two different groups (e.g., sex), and paired *t*-tests were performed to examine the difference between pre- and post-training values. Pearson correlations were used to examine associations between nutrient intakes and RET responses (changes in lean mass and 1RM) and to identify potential covariates for further analyses. 

The differences between choline groups were analyzed by one-way ANOVA. The assumption of equal variance was checked using Levene’s test, and the Tukey method was used to perform pairwise comparisons. When the equal variance assumption was not met, Welch’s variance-weighted ANOVA test was performed. ANCOVA tests were conducted to examine the effects of dietary choline, controlling for effects of potential confounders (e.g., cholesterol and other nutrients, sex, age, lean mass, etc.) on RET responses. 

Multiple linear regression analyses were performed to examine the independent effects of choline consumption and any other factors on RET responses. Composite strength was defined as chest press 1RM + leg press 1RM, and percent change was calculated as 100 × (post-training measurement − pre-training measurement)/pre-training measurement. The *p*-values < 0.05 were considered statistically significant, and data are presented as mean ± SD, unless stated otherwise.

## 3. Results

### 3.1. Demographics

The baseline characteristics of the 37 participants who completed the 12 weeks of RET are presented in Table 1. There were no differences between choline groups in age, male-to-female ratio, height, weight, lean mass, or adiposity.

### 3.2. Nutritional Compliance

Participants successfully followed the dietary guidelines of the study and met all the requirements for nutritional intake. On average, the participants consumed 27 kcal/kg/d of energy, 3.2 g/kg/d of carbohydrate, 1.4 g/kg/d of protein, and 1.0 g/kg/d of fat throughout the study, and there was no difference in nutrient intake between choline groups, except for cholesterol consumption (Table 2).

The mean choline consumption was 6.2 ± 1.2 mg/kg lean/d for the low group (~51% of AI), 8.1 ± 1.6 mg/kg lean/d for the med group (~68% of AI), and 14.2 ± 3.0 mg/kg lean/d for the high group (~118% of AI). The choline intake from participants’ own diets was 5.9 ± 2.2 mg/kg lean/d (low = 5.5 ± 1.2 mg/kg lean/d, med = 5.3 ± 1.6 mg/kg lean/d, and high = 6.7 ± 3.0 mg/kg lean/d, *p* = 0.20), and the supplement provided additional 0.7, 2.8, and 7.5 mg/kg lean/d of choline for low, med, and high groups, respectively. Choline intake (mg/kg lean/d) was correlated with folate (DFE/kg lean/d; r = 0.58; *p* < 0.001), vitamin B_5_ (mg/kg lean/d; r = 0.52; *p* = 0.001), vitamin B_6_ (mg/kg lean/d; r = 0.53; *p* < 0.001), vitamin B_12_ (mcg/kg lean/d; r = 0.59; *p* < 0.001), and cholesterol (mg/kg lean/d; r = 0.88; *p* < 0.001).

### 3.3. RET Responses

RET resulted in significant increases in lean mass and strength from baseline in all three groups while only low and med groups lost significant body fat (Table 3, Table 4 and Table 5). However, there was no difference between groups in changes in lean or fat mass (Table 3 and Table 4).

Because there was no difference between males and females in muscle responses to RET (male vs. female; percent change in lean mass: 3.6 ± 2.0 vs. 3.6 ± 3.6, *p* = 0.99; percent change in composite strength: 41.0 ± 38.8 vs. 33.7 ± 23.5, *p* = 0.52), data were pooled for further analyses. Observed correlation coefficients (r) of percent change in composite strength were 0.29 (*p* = 0.097) with choline (mg/kg lean/s) and 0.31 (*p* = 0.07) with betaine (mg/kg lean/s). Percent change in lean mass was significantly correlated with vitamin B_5_ intake (mg/kg lean/d; r = 0.35; *p* = 0.04), while choline consumption was not significantly correlated with lean mass gains with RET (r = 0.25; *p* = 0.14). 

Because choline consumption was significantly correlated with folate, vitamin B_5_, vitamin B_6_, vitamin B_12_, and cholesterol, and RET responses were associated with betaine and vitamin B_5_, ANCOVA tests using these and other potential confounders, including age, lean mass, and other major dietary factors (e.g., protein intake), as covariates were conducted. The results showed a significant difference in percent change in composite strength between choline groups (Figure 3; *p* = 0.034). The low choline group showed reduced composite strength gain (%) when compared with the med (*p* = 0.035) or high (*p* = 0.085) choline groups (low = 19.4 ± 8.2%, med = 46.8 ± 8.9%, and high = 47.4 ± 8.1%), after adjusting for covariates. The covariates appearing in the final model were lean mass, protein, betaine, and vitamin B_12_. The other potential confounders were removed during the model selection/simplification processes due to their insignificant contribution to variability of percent gains in composite strength.

Since RET responses in med and high groups were similar (Figure 3 and Table 4), additional analyses were conducted after the med and high groups were pooled. Independent *t*-test results showed significant differences between low and med–high groups in changes (%) in leg press 1RM (low = 25.3 ± 24.0 vs. med–high = 48.2 ± 42.8, *p* < 0.05) and composite strength (low = 129.9 ± 84.2 vs. med–high = 182.1 ± 147.3, *p* = 0.05). ANCOVA test using lean mass (kg), protein (g/kg lean/d), betaine (mg/kg lean/d), and vitamin B_12_ (mcg/kg lean/d) as covariates also showed that low group had significantly lower thigh-muscle quality–strength (TMQ-S) improvements compared with med–high group, while the differences in leg press and composite peak power were not statistically significant (Table 6).

Multiple linear regression analyses were also conducted to evaluate the independent association of dietary choline and other nutrients, as well as potential confounders. All the variables were initially entered into the regression equation, and the variables were sequentially removed at each step with the backward elimination method. The final model showed that low choline intake independently predicted percent change in composite strength, with betaine intake, male sex, and lean mass remaining in the model (adjusted R^2^ = 0.215; *p* = 0.02; Table 7). Even though cholesterol was previously shown to be associated with lean mass and strength gains [30], it did not predict the variability of strength gains in the present study.

### 3.4. Blood Lipids and Liver Damage Markers 

Since choline deficiency causes liver/muscle damage and altered lipoprotein/blood lipid metabolism, blood markers for liver damage (alanine aminotransferase (ALT) and aspartate aminotransferase (AST)), muscle damage (creatine kinase (CK)), and blood lipid profiles (triacylglycerol (TAG), total cholesterol, high-density-lipoprotein cholesterol (HDL-C), and low-density-lipoprotein cholesterol (LDL-C)) were also measured. The results showed no effect of choline intake on any of these clinical blood markers (Table 8).

## 4. Discussion

The purpose of the present study was to determine the effects of different amounts of choline intakes on muscle responses to RET. We found that low choline intake (~51% of AI) resulted in diminished strength gains with 12 weeks of RET in 50-to-69-year-old individuals, compared to choline intakes of ~68% or ~118% of AI. We also observed that a high choline intake (greater than AI) did not provide additional positive effects on RET responses.

It is well known that RET is important for the health and well-being of humans, especially for older individuals [31,32]. It is also well established that muscle responses to RET can be influenced by nutrition [33]. There have been many studies that showed the effects of nutritional interventions on muscle responses to RET or exercise performances, especially the effects of nutrients including protein, amino acids, or vitamins [33,34,35]. Because choline is an essential nutrient and plays important roles in human physiology, we hypothesized that insufficient choline intake would negatively affect muscle responses to RET. 

Choline may affect muscle responses to RET through many potential mechanisms. Since choline is a precursor to a neurotransmitter, ACh, which relays a signal from motor neurons to skeletal muscle to contract and generate force, and of which synthesis is reported to be affected by availability of choline [36,37], insufficient choline consumption may limit the availability of ACh at the neuromuscular junction (NMJ) and, in turn, muscle contraction and force generation. ACh is synthesized in cholinergic nerves by choline acetyltransferase (ChAT), using acetyl-CoA and choline as substrates. After it is released into the synaptic cleft and binds to ACh receptors on the muscle cell membrane, it is broken down to acetate and choline by acetylcholinesterase. Choline is then taken up by choline transporter proteins on nerve cells and recycled to resynthesize ACh by ChAT. Crockett et al. [38] reported that ChAT activity is inversely related to the size of the muscle fiber and positively related to the resistance of the muscle fiber to fatigue. Since fast twitch (FT) muscle fibers are generally larger in size and less resistant to fatigue, ChAT activity may be lower in FT fibers, and FT fibers may have less ability to recycle choline and resynthesize ACh, making them more sensitive to and reliant on choline supplied by the circulation (eventually from diet), compared with slow twitch (ST) fibers. Since FT fibers are mostly recruited and utilized with RE, if choline affects RET responses via ACh, this may explain, at least in part, why choline intake may influence muscle responses to endurance exercise and RE differently. Moreover, Herscovich and Gershon [39] reported that aging decreases activities of ChAT, suggesting that the importance of sufficient choline intake may be amplified in the older population. 

Choline can also influence methylation reactions. A portion of choline is irreversibly converted to betaine, which is used to convert homocysteine to methionine, which is then used to generate *S*-adenosyl methionine (SAM), a universal methyl group donor. Therefore, choline and its metabolite betaine may affect methylation, which plays crucial roles in lipid synthesis, the epigenetic control of gene expression/protein synthesis, and the regulation of many other metabolic pathways [40,41]. Since betaine, via SAM, may contribute to the synthesis of creatine and is an important osmolyte maintaining fluid balance, studies have been conducted to examine the effect of betaine ingestion on exercise performance and body composition. The results generally showed positive effects of betaine on performance and body composition, with some inconsistencies [11,42]. For example, body composition was favorably affected, and training volume was increased with six weeks of betaine supplementation in the study of Cholewa et al. [43], whereas 10 d of betaine supplementation did not affect muscle creatine content, or 1RM/power of bench press/squat in another study [12]. In the present study, together with choline, betaine was independently associated with change (%) in composite strength, suggesting that multiple mechanisms are at work. Future studies are warranted to investigate the independent roles that choline and betaine each plays on RET responses.

Choline is also a precursor to PC, the predominant type of phospholipid in all cell membranes. Therefore, choline contributes to the stability and integrity of cell membranes, and choline deficiency results in weakened cell membranes and the leakage of intracellular enzymes into the circulation. Da Costa et al. [10] reported that blood concentrations of CK, a muscle-cell-damage marker, increased up to 66-fold with a severely low (<10% of AI) choline intake. Insufficient choline consumption and the resultant decrease in the cytidine diphosphocholine pathway (which makes PC from dietary choline) can also lead to the perturbation of PC homeostasis and induce cell death [44]. Therefore, choline-deficient diets may result in a weak and compromised cell membrane/structure, which may negatively affect the ability of skeletal muscle to withstand mechanical stresses imposed by exercise, especially during RET. 

We did not observe any negative effects of low choline consumption on plasma CK concentrations. This may be explained by the difference in the amount of choline intakes between da Costa et al.’s study (<10% of AI) [10] and the present study (low group: ~50% of AI). It appears that only extremely low choline intake may induce muscle-cell-membrane damage and the leakage of CK into the circulation. Similarly, there was no effect of low choline intake on blood concentrations of ALT and AST, also suggesting that only severely deficient choline consumption may result in the release of those enzymes. However, hepatic steatosis was observed in a previous study [45] without leakage of ALT or AST in many subjects in response to a choline-deficient diet, suggesting that liver dysfunction can occur even without increases in intracellular liver enzyme concentrations in the blood. It remains to be determined whether a moderately low choline intake (as in the present study) can subtly compromise and weaken membrane integrity, thereby negatively affecting muscle functions, while still not allowing leakage of intracellular enzymes into the circulation.

We also examined the effect of choline intake on blood lipid profiles, since total blood cholesterol and LDL-C were previously reported to be positively associated with lean mass gains [30], and dietary choline influences the synthesis of lipoproteins that transport cholesterol and fat in the circulation [1]. However, there was no effect of choline consumption on any of the blood lipids and lipoproteins, indicating that a moderately low choline intake may not negatively affect blood lipid profiles. In addition, while dietary cholesterol was previously shown to be associated with lean mass and strength gains with RET in older adults [30], it did not contribute much to the variability of strength gains in the present study. The reason for this discrepancy is unclear, but it can be speculated that the results of the previous study may have been confounded by the inability to separate the effect of choline from that of cholesterol because information on the choline content of foods was not widely available when the previous study was conducted. Since many choline-rich foods are also rich in cholesterol, future studies may be needed to examine the independent roles that choline and cholesterol each plays on muscle responses to RET.

It should be noted that our study results do not necessarily promote high consumption of choline. While the lower (~51% of AI) choline intake resulted in less strength gains, the higher (~118% of AI) choline consumption did not provide any additional training benefits in the present study. Therefore, more emphasis should be placed upon consuming an adequate amount of choline rather than encouraging a higher-than-necessary amount of choline intake. Choline is mainly found in animal-based foods such as eggs and meats, which generally contain high cholesterol [29]. If one is concerned about cholesterol in those choline rich foods, they can consume alternative food items that are high in choline but low in cholesterol, such as soybeans, fish, potatoes, mushrooms, and cruciferous vegetables [29]. However, there is no clear association between egg consumption and cardiovascular disease mortality [46], and some studies found that egg consumption may even improve lipoprotein profiles [47] or suppress ischemic heart diseases [48]. According to a recent study, consuming 7–14 eggs/week as part of a balanced diet can be beneficial for most people [49]. 

Our study has some limitations, including the inability to determine the mechanisms through which choline may affect RET responses. Since we consistently observed the effects of dietary choline (especially the negative effects of low choline intake [~50% of AI]) on muscle responses to RET, future studies should be focused on elucidating the mechanism(s) of those effects. Moreover, the well-known tendency to under-report food intake associated with diet logs may have obscured the accuracy of our data [50]. However, we believe that, compared with the 3-day food records commonly used in nutrition studies or the 7-day weighed food records which are considered to be the best method currently, the >48-day food records (≥4 d of diet records/week for the entire 12 weeks) we required from our participants minimized the issues related to the inaccurate reporting of food intake. We also expect that errors associated with this method would be consistent across all the choline groups, having little possibility of altering the overall conclusions of the present study.

## 5. Conclusions

This randomized placebo-controlled trial provides additional evidence that a lower intake of choline (~51% of AI) results in significantly diminished strength gains in response to 12 weeks of RET compared with higher choline intakes of 68% or 118% of AI in 50-to-69-year-old individuals; however, there was no effect of dietary choline on lean mass gains. The consistency of this effect at about 50% of AI is particularly significant because as much as 40% of the older population is consuming this low level of choline where there are no overt clinical signs of deficiency and considering the potential effects of choline on the devastating effects of age-associated loss of muscle function.

## Figures and Tables

**Figure 1 nutrients-15-03874-f001:**
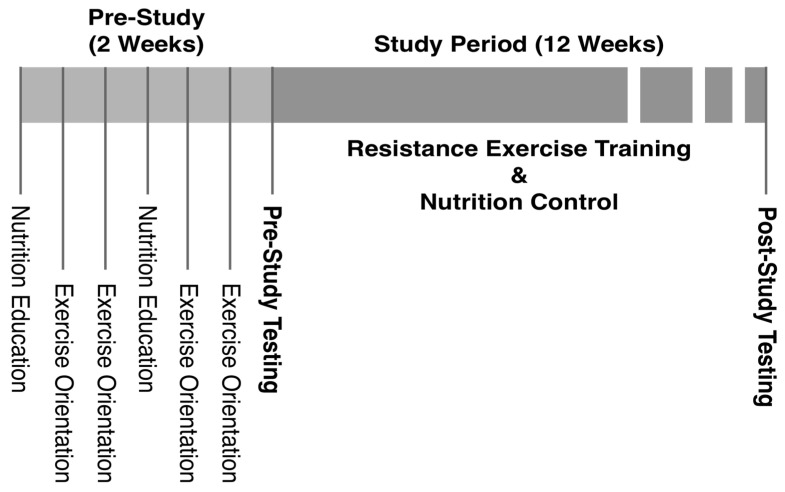
Study timeline.

**Figure 2 nutrients-15-03874-f002:**
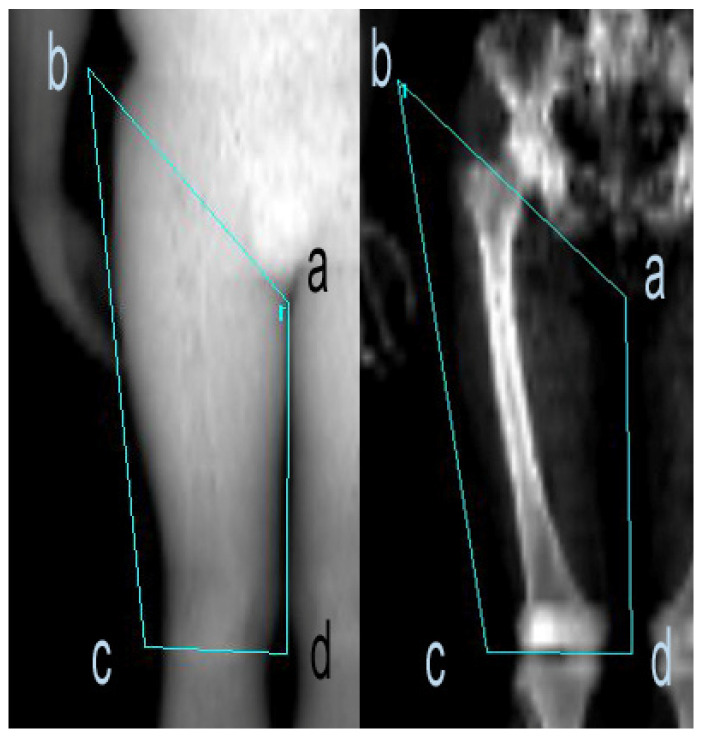
Example of DEXA scan image of right thigh. A four-sided polygon was constructed with one point below the pubic bone (a) that was connected to another point (b) as to obliquely cross the intertrochanteric crest of the femur bone. Points (c) and (d) were positioned as to be traversing the tibiofemoral joint. Points (b)–(c) and (a)–(d) were connected to ensure that the entire thigh was encompassed.

**Figure 3 nutrients-15-03874-f003:**
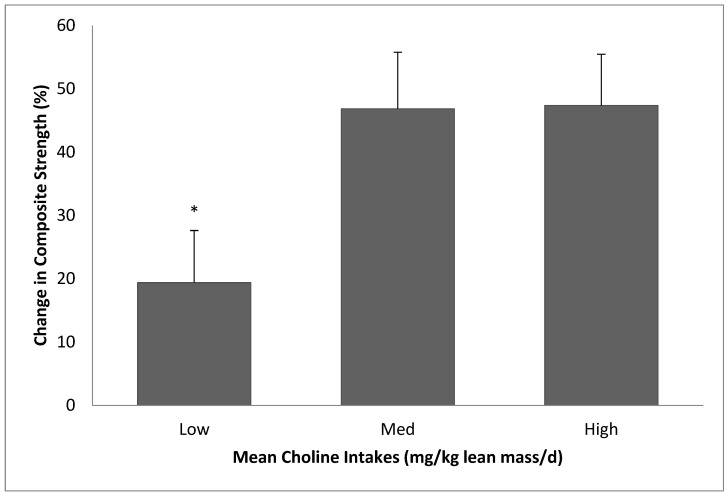
Changes in composite strength as a function of choline intake. Data are presented as least squares mean ± standard error (SE). Low: choline intake of 6.2 ± 1.2 mg/kg lean mass/d. Med: choline intake of 8.1 ± 1.6 mg/kg lean/d. High: choline intake of 14.2 ± 3.0 mg/kg lean/d. Low group gained less strength when compared with med (*p* = 0.035) or high (*p* = 0.085) groups. Composite strength is defined as chest press 1RM + leg press 1RM. Covariates appearing in the model: lean mass (kg), protein (g/kg lean/d), betaine (mg/kg lean/d), and vitamin B_12_ (mcg/kg lean/d). * Significant difference.

**Table 1 nutrients-15-03874-t001:** Participant’s baseline characteristics ^1^.

	Low (n = 13)	Med (n = 11)	High (n = 13)	*p*-Values
Age (years)	58.9 ± 7.3	60.7 ± 4.6	60.0 ± 4.5	0.78
Male/female	6/7	5/6	4/9	0.69
Height (cm)	170.7 ± 9.7	168.9 ± 8.9	166.4 ± 7.6	0.47
Weight (kg)	83.5 ± 14.9	80.8 ± 22.9	78.5 ± 15.9	0.77
Body fat (kg)	31.3 ± 12.2	27.7 ± 10.7	31.3 ± 7.9	0.65
Lean mass (kg)	48.8 ± 9.7	44.6 ± 8.1	44.0 ± 11.3	0.42
BMI (kg/m^2^)	28.8 ± 5.8	28.1 ± 5.7	28.2 ± 4.7	0.94

^1^ Data are presented as mean ± SD. Low: choline intake of 6.2 ± 1.2 mg/kg lean mass/d. Med: choline intake of 8.1 ± 1.6 mg/kg lean/d. High: choline intake of 14.2 ± 3.0 mg/kg lean/d. No differences were observed between choline groups.

**Table 2 nutrients-15-03874-t002:** Intake of major nutrients and vitamins ^1^.

	Low (n = 13)	Med (n = 11)	High (n = 13)	*p*-Values
Total energy (kcal/kg/d)	27.7 ± 5.8	25.6 ± 6.3	27.7 ± 6.9	0.58
Carbohydrate (g/kg/d)	3.4 ± 0.9	3.0 ± 0.7	3.3 ± 0.8	0.39
Protein (g/kg/d)	1.4 ± 0.2	1.4 ± 0.3	1.4 ± 0.2	0.45
Fat (g/kg/d)	1.0 ± 0.4	0.9 ± 0.2	1.0 ± 0.4	0.36
% kcal from carbohydrate ^2^	49.6 ± 4.3	47.9 ± 2.7	49.5 ± 4.9	0.58
% kcal from protein ^2^	16.9 ± 3.0	18.4 ± 1.4	17.1 ± 2.3	0.25
% kcal from fat ^2^	31.7 ± 4.5	31.5 ± 2.1	30.8 ± 5.8	0.86
Folate (DFE/kg/d)	5.8 ± 1.6	6.3 ± 2.8	7.3 ± 2.1	0.21
Vitamin B_5_ (mg/kg/d)	0.07 ±0.02	0.08 ± 0.05	0.09 ± 0.02	0.33
Vitamin B_6_ (mg/kg/d)	0.03 ± 0.01	0.03 ± 0.01	0.03 ± 0.01	0.35
Vitamin B_12_ (mcg/kg/d)	0.06 ± 0.03	0.07 ± 0.04	0.08 ± 0.03	0.16
Betaine (mg/kg/d)	0.7 ± 0.5	0.9 ± 1.1	1.0 ± 0.9	0.61
Cholesterol (mg/kg/d)	1.9 ± 0.4	4.0 ± 0.7	7.7 ± 1.1	<0.001

^1^ Data are presented as mean ± SD and include values from supplement. Low: choline intake of 6.2 ± 1.2 mg/kg lean mass/d. Med: choline intake of 8.1 ± 1.6 mg/kg lean/d. High: choline intake of 14.2 ± 3.0 mg/kg lean/d. ^2^ Values from participants’ diet only. No differences were observed between the choline groups, except for cholesterol intake.

**Table 3 nutrients-15-03874-t003:** Changes in body composition as affected by choline intake ^1^.

	Low (n = 13)	Med (n = 11)	High (n = 13)	*p*-Values
Change in lean mass (kg)	1.6 ± 1.5	1.6 ± 1.5	1.7 ± 0.9	0.95
Percent change in lean mass	2.9 ± 3.3	3.9 ± 3.7	4.1 ± 2.3	0.58
Change in body fat (kg)	−0.8 ± 1.4	−1.2 ± 1.1	−0.6 ± 1.8	0.75
Percent change in body fat	−3.4 ± 5.8	−4.8 ± 5.0	−2.0 ± 6.7	0.55

^1^ Data are presented as mean ± SD. Low: choline intake of 6.2 ± 1.2 mg/kg lean mass/d. Med: choline intake of 8.1 ± 1.6 mg/kg lean/d. High: choline intake of 14.2 ± 3.0 mg/kg lean/d. All changes from baseline are statistically significant, except for the change in body fat in the high group. No significant differences were observed between groups.

**Table 4 nutrients-15-03874-t004:** Changes (%) in 1RM, as affected by choline intake ^1^.

	Low (n = 13)	Med (n = 11)	High (n = 13)	*p*-Values
Leg press	25.3 ± 24.0	45.7 ± 40.2	50.3 ± 46.4	0.22
Chest press	25.4 ± 13.4	23.0 ± 15.1	31.0 ± 24.0	0.55
Lat pull down	17.2 ± 13.3	23.1 ± 26.7	30.6 ± 23.7	0.30
Knee extension	21.7 ± 23.0	14.8 ± 30.3	37.5 ± 44.9	0.27
Leg curls	20.9 ± 19.9	35.6 ± 25.0	36.0 ± 26.6	0.21
Triceps extension	26.8 ± 17.6	31.0 ± 21.2	36.8 ± 27.5	0.53
Biceps curls	41.4 ± 52.8	38.2 ± 26.0	36.8 ± 39.8	0.96

^1^ Data are presented as mean ± SD. Low: choline intake of 6.2 ± 1.2 mg/kg lean mass/d. Med: choline intake of 8.1 ± 1.6 mg/kg lean/d. High: choline intake of 14.2 ± 3.0 mg/kg lean/d. All changes from baseline are statistically significant, but no significant differences were observed between groups.

**Table 5 nutrients-15-03874-t005:** Pre- and post-RET 1RM values (kg) ^1^.

		Low (n = 13)	Med (n = 11)	High (n = 13)	*p*-Values
Leg press	Pre	232.1 ± 85.8 *	183.1 ± 38.7	164.6 ± 65.4	0.04
	Post	281.8 ± 83.2	256.9 ± 51.1	240.4 ± 95.1	0.42
Chest press	Pre	36.6 ± 15.4	33.7 ± 13.4	32.9 ± 20.5	0.84
	Post	45.8 ± 19.2	40.3 ± 13.3	41.6 ± 24.0	0.77
Lat pull down	Pre	56.9 ± 21.4	51.6 ± 13.2	44.5 ± 18.0	0.23
	Post	65.2 ± 20.9	62.6 ± 16.1	56.8 ± 22.0	0.56
Knee extension	Pre	45.3 ± 20.8	41.3 ± 15.5	36.4 ± 19.6	0.49
	Post	52.9 ± 20.3	49.7 ± 12.4	45.8 ± 20.4	0.62
Leg curls	Pre	59.1 ± 18.2	55.3 ± 15.6	44.5 ± 17.2	0.09
	Post	70.3 ± 18.9	73.8 ± 19.1	59.1 ± 21.6	0.18
Triceps extension	Pre	78.2 ± 22.1	68.5 ± 15.9	65.6 ± 30.6	0.39
	Post	99.4 ± 32.7	88.5 ± 19.8	87.0 ± 38.3	0.57
Biceps curls	Pre	16.6 ± 8.7	16.1 ± 8.7	13.8 ± 10.0	0.71
	Post	20.9 ± 8.7	20.8 ± 8.8	17.1 ± 9.6	0.49

^1^ Data are presented as mean ± SD. Low: choline intake of 6.2 ± 1.2 mg/kg lean mass/d. Med: choline intake of 8.1 ± 1.6 mg/kg lean/d. High: choline intake of 14.2 ± 3.0 mg/kg lean/d. * Significant difference from high group (*p* < 0.05).

**Table 6 nutrients-15-03874-t006:** Changes (%) in peak power and thigh-muscle quality, as affected by choline intake ^1^.

	Low (n = 13)	Med–High (n = 24)	*p*-Values
Leg press peak power (%Δ in W)	19.0 ± 13.5	30.0 ± 22.4	0.072
Composite peak power (%Δ in W)	17.4 ± 12.5	25.5 ± 17.3	0.110
TMQ-S ^2^	12.3 ± 9.6	46.4 ± 7.0	0.010

^1^ Data are presented as mean ± SD. Composite peak power = leg press peak power + chest press peak power. TMQ-S = leg press 1RM (kg)/total thigh lean mass for both legs (kg). All changes from baseline were statistically significant. ^2^ ANCOVA analysis results presented as least-squares mean ± standard error (SE). Covariates appearing in the model: lean mass (kg), protein (g/kg lean/d), betaine (mg/kg lean/d), and vitamin B_12_ (mcg/kg lean/d).

**Table 7 nutrients-15-03874-t007:** Multiple regression analysis of the independent effects of low choline intake, lean mass, and betaine intake on change (%) in composite strength following 12 weeks of RET ^1^.

	Unstandardized Coefficients		
Predictors	Β	SE	*p*-Values
(Intercept)	0.86	1.04	0.42
Low choline intake *	−0.62	0.27	0.03
Male sex	−0.89	0.50	0.09
Lean mass (kg) *	0.00006	0.00003	0.03
Betaine intake (mg/kg/d) *	0.23	0.11	0.04

^1^ SE: standard error. Low choline intake: choline consumption of 6.2 ± 1.2 mg/kg lean mass/d. * Statistical significance (*p* < 0.05).

**Table 8 nutrients-15-03874-t008:** Pre- and post-RET values of select blood lipids and enzymes as affected by choline intake ^1^.

		Low (n = 10) ^2^	Med (n = 10) ^2^	High (n = 10) ^2^	*p*-Values
ALT (U/L)	Pre	42.3 ± 8.6	32.6 ± 9.6	39.4 ± 24.9	0.36
	Post	36.5 ± 8.7	35.6 ± 14.9	37.2 ± 12.1	0.96
AST (U/L)	Pre	38.4 ± 11.1	28.7 ± 5.4	35.0 ± 16.5	0.17
	Post	33.4 ± 7.8	32.9 ± 9.2	31.2 ± 9.2	0.83
Total cholesterol (mg/dL)	Pre	184.4 ± 25.7	193.9 ± 20.3	181.2 ± 31.3	0.53 0.08
	Post	176.8 ± 15.1	206.3 ± 29.8	191.6 ± 36.4	
HDL-C (mg/dL)	Pre	55.8 ± 10.7	55.2 ± 10.4	52.8 ± 12.4	0.81
	Post	53.2 ± 9.2	54.8 ± 9.5	53.9 ± 12.8	0.94
LDL-C (mg/dL)	Pre	107.8 ± 20.6	115.3 ± 20.1	104.0 ± 30.6	0.57
	Post	104.1 ± 16.2	127.0 ± 31.1	111.6 ± 33.4	0.18
TAG (mg/dL)	Pre	103.8 ± 34.6	117.1 ± 48.9	122.3 ± 78.6	0.73
	Post	97.1 ± 37.5	122.2 ± 62.0	131.4 ± 80.0	0.42
CK (U/L) ^2^	Pre	119.0 ± 50.4	98.7 ± 47.9	128.1 ± 78.9	0.70
	Post	77.3 ± 25.4	85.8 ± 41.4	99.3 ± 29.3	0.54

^1^ Data are presented as mean ± SD. ^2^ Sample sizes were reduced due to blood sample availability (n is for CK: low = 4, med = 6, and high = 8; CK was measured in blood samples taken 48 h after the first and last exercise sessions of RET). Low: choline intake of 6.2 ± 1.2 mg/kg lean mass/d. Med: choline intake of 8.1 ± 1.6 mg/kg lean/d. High: choline intake of 14.2 ± 3.0 mg/kg lean/d. ALT: alanine aminotransferase. AST: aspartate aminotransferase. HDL-C: high-density-lipoprotein cholesterol. LDL-C: low-density-lipoprotein cholesterol. TAG: triacylglycerol. CK: creatine kinase. There was no difference between or within groups in any of the blood-marker concentrations.

## Data Availability

The data presented in this study are available from the authors upon reasonable request.

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
