# Peer review of "The Effect of Choline and Resistance Training on Strength and Lean Mass in Older Adults"

_nutrients, 2023, doi:10.3390/nu15183874_

Round 1

Reviewer 1 Report

Comments to the Author:

I thank to the editors for the opportunity to review this study, beside I would also like to congratulate the authors for the made effort in their study. The present manuscript by Chang Woock et al., analyzed “The Effect of Choline and Resistance Training on Strength and Lean Mass in Older Adults”. The authors determine the effects of various amounts of choline intakes (including higher than AI) on muscle responses to resistance exercise training. The currently the paper needs a lot of information and clarification of many doubts in the experimental design.

1.      First, the first sentence of the introduction must be supported by a reference. In addition, the authors say "However, the amount of choline produced via de novo synthesis is not sufficient to support total choline requirement". This argument must be supported by several studies, as it is the main reason for the initiation of their study. From line 42-48 there are no pauses, the paragraph is too long, it may confuse the reader.

2.      The authors of this study have finally used a number of 37 healthy males and females; however, how did the authors know that with 37 subjects the statistical power of the study was sufficient?  I recommend that you use the G-Power program; it will be very useful and provide you with valuable information for your study.

3.      In addition, I do not know if the present study is in compliance with the Declaration of Helsinki. Please clarify this issue.

4.      Could the authors justify the 12 weeks of resistance training and the 3 sessions per week? Why 12 weeks and not more or less? In addition, it should justify why the exercises that were used were chosen. What were the real reasons for using these strength exercises? Add this justification to the main paper.

5.      The design of the resistance training programme should be better justified; in fact the authors have not used any reference to support such a design. Everything in experimental design must be supported by references and you have not used any references. The same applies to the design of nutritional monitoring; everything must be backed up by references, which guarantee that the method that has been carried out is correct.

6.      It would be interesting to add the 1RM data for each exercise, and to add the before and after strength data.

7.      The first paragraph of the discussion should be much better structured. First the main aim of the study and then the most relevant results. It is not necessary to add more information because it confuses the reader.

Reviewer 2 Report

Concerning the manuscript: The Effect of Choline and Resistance Training on Strength and Lean Mass in Older Adults, submitted to Nutrients. It is a good work with sound methodology and interesting results. Overall, the paper will contribute to knowledge. With the utmost respect, allow me to give you a few suggestions.

Title & Abstract

Do the title and abstract cover the main aspect of the work?

This is an interesting study with new information which can be very useful, however a brief explanation about the functions and proprieties of choline could be more highlighted in the abstract of manuscript.

We observed low choline intake was negatively associated with strength and lean mass gains following 12 weeks”. This sentence must be revised.

Introduction

Does the introduction provide background and information relevant to the study?

·  The reasons for studying strength, lean mass could be more highlighted. I suggest the inclusion of a more detailed physiological explaining how choline could help muscle performance (for example cell membrane integrity would be more preserved). How acetylcholine, lipoprotein synthesis or betaine could affect strength/muscle performance? These physiological pathways underlying strength/muscle performance were not valued in introduction.

·  Similarly, the different dosages could be more highlighted in the objectives: Line 61 “The purpose of the present study was to determine the effects of various amounts of choline intakes (including higher than AI).”

Material and Methods

Are the methods clear and replicable? Do all the results presented match the methods described?

·  The experimental design seems proper, but more technical details about the procedures are needed. It is important to describe when data were collected (time of day, environmental condition, season of data collection, data collection duration). Were researchers trained or had training in conducting tests?

·  Didactics would improve with the inclusion of a figure (some scheme drawn by the authors) explaining timeline of experiments. Please, change to be clearer for the reader.

·  The authors should report some pictures demonstrating the conduction of experiments (1RM, peak power, body composition, resting metabolic rate (RMR) and blood tests). This could be added in a supplementary file.

·  Why other exercises (Lat Pull Down, Knee Extension, Leg Curls, Triceps Extension, Biceps Curls) were not used in the obtention of composite strength (ONLY chest press 1RM + leg press 1RM were used). This must be referenced.

·  There is mention (in line 185) that student’s independent t-tests were used to compare means of two different groups (e.g., sex), but I have no found these analyses.

Results

If relevant are the results novel? Does the study provide an advance in the field? Is the data plausible?

·  I suggest the inclusion of Pearson’s correlations for understanding physiological interactions (for example strength vs choline intake and others). This could be made using data from all 37 individuals (for amplifying the sample amount). If using figures, I suggest that you accurately discriminate the groups using different symbols or colors. For example:

Low group (0.7 mg of choline) > blue square  

Med group (2.8 mg of choline) > green triangle

High group (7.5 mg of choline) > red circle

·  Regarding data exploration (table 3), why only absolute and percentual changes were analyzed ? Data could be more highlighted by using raw values for lean mass (kg and kg/body mass) and body fat kg and kg/body mass). The same is true for the table 4 and 5.

·  In table 7, I presume that p values refer to comparisons among choline intake groups. My doubt is whether pre vs post comparisons were performed.

Discussion

Do the findings described by the author correlate with the results? Are the findings relevant?

·  The topic is relevant and seems proper. Considering that, in the present study, there is a concern for the improvement of physical performance. Some issues could be raised. I suggest the citation of studies in which nutritional intervention were explored in practitioners of resistance training. I would like to know the authors' line of reasoning about this.

·  It is possible to note that high group showed the biggest changes for cholesterol when compared to other groups. Could this be a confounding factor ? Which the possible consequences of elevated cholesterol on performance? I recommend more emphasis on discussing their findings.

•The authors should make a recommendation about the unrestrained use of egg yolk: Considering the importance of choline in the context of strength performance, caution should be established for preventing alterations in cholesterol. The authors should suggest other nutritional alternatives (which also contain choline) to help practitioners of resistance training.  

Round 2

Reviewer 1 Report

For the author:

I appreciate authors’ effort. The authors have obviously spent considerable time revising the manuscript and their hard work is clearly paying off. This manuscript is drastically improved from the original submission. The message is very clear, the language is much more clean, and the issues in the first version were corrected. Besides, the authors have answered all my comments successfully. For this reason, I encourage to editor to consider this manuscript for publication for the interesting value of the study realized, that now it is a much more robust study.